# Development of the Active Ageing Awareness Questionnaire in Malaysia

**DOI:** 10.3390/healthcare9050499

**Published:** 2021-04-22

**Authors:** Nor Hana Ahmad Bahuri, Hussein Rizal, Hazreen Abdul Majid, Mas Ayu Said, Tin Tin Su

**Affiliations:** 1Department of Social and Preventive Medicine, Faculty of Medicine, University of Malaya, Kuala Lumpur 50603, Malaysia; drhannadrph@gmail.com; 2Centre for Population Health (CePH), Department of Social and Preventive Medicine, Faculty of Medicine, University of Malaya, Kuala Lumpur 50603, Malaysia; husseinriz@um.edu.my (H.R.); hazreen@ummc.edu.my (H.A.M.); 3Centre for Epidemiology and Evidence-Based Practice, Department of Social and Preventive Medicine, Faculty of Medicine, University of Malaya, Kuala Lumpur 50603, Malaysia; 4South East Asia Community Observatory (SEACO), Jeffrey Cheah School of Medicine & Health Sciences, Monash University, Bandar Sunway, Subang Jaya 47500, Malaysia; TinTin.Su@monash.edu

**Keywords:** active ageing, confirmatory factor analysis, exploratory factor analysis, adults, quality of life

## Abstract

The world’s ageing population is associated with increased morbidity, disability, and social and financial insecurity, which may affect quality of life (QoL). Therefore, the World Health Organization (WHO) endorsed the Active Ageing Framework (AAF) in 2002 to enhance QoL as people age. However, little is known about the status of awareness of active ageing among the population, and there is no appropriate tool for assessment. Hence, the Awareness of Active Ageing Questionnaire (AAAQ) was developed. The content, linguistic and face validations together with test-retest reliability were conducted. Exploratory factor analysis (EFA) and confirmatory factor analysis (CFA) were performed to test the structural validity of the AAAQ. A total of 110 participants (mean ± SD = 50.19 ± 5.52) were selected for the pilot, 81 participants (mean ± SD = 49.40 ± 5.70) for the test-retest, and 404 participants (mean ± SD = 49.90 ± 5.80) for CFA and EFA tests. The 16-item AAAQ Malay version showed satisfactory reliability and validity. The Cronbach’s alpha was more than 0.7 and showed good fit: Cmin/df = 2.771, GFI = 0.903, TLI = 0.951, RMSEA = 0.08. The AAAQ is suitable for measuring the awareness of active ageing among the middle-aged population in Malaysia.

## 1. Introduction

In the year 2050, the global ageing population will outnumber the population aged less than 15 years old for the first time [1]. It has been estimated that Malaysia will become an aged nation in 2030 [2], and the proportion of older persons will be at least 15 percent of the total population [3]. In general, an increasing ageing population reflects the tremendous achievements of public health policies and social and economic development. However, this phenomenon has resulted in profound health, social, and economic implications and led to significant challenges. Longer human lives have led to a global burden of late-life disease [4]. The older population suffers the highest rate of disability in areas such as hearing [5], vision [6], memory loss [7], urinary incontinence [8], and joint pain [9].

This phenomenon has resulted in the endorsement by the World Health Organization (WHO) of the Active Ageing Framework in 2002 [10], which serves as a guide for policymakers for designing policies and programs that aim to ensure the quality of life (QoL) of older persons. Active ageing is defined as “the process of optimising the opportunity of health, participation and security in order to enhance the QoL as people age” [10]. Whereas QoL is defined as “an individual’s perception of their position in their life in the context of the culture and value systems in which they live and in relation to their goals, expectations, standards and concerns” [11].

Studies have found that the perceived QoL of current older Malaysians (age 60 and above) is poor [12,13], and that older women have significantly lower psychological well-being as compared to their counterparts [14]. In addition, it was reported that the prevalence of successful ageing among older Malaysians is only 13.8 percent. Successful ageing is described as “low probability of disease and disease-related disability; high cognitive and physical functioning and active engagement in life” [15]. Aside from sociodemographic, factors such as differences in socioeconomic and medical illness status have been shown to be associated with health-related QoL [16]. Therefore, there is a need to promote active ageing among the adult population through intervention programs that assist them to age actively.

There is currently no specific and standardised tool available to measure awareness of active ageing. Several studies have been conducted on awareness of ageing in many settings internationally; however, the aspects of ‘participation’ and ‘security’ dimensions were left out in the questionnaires [17,18]. In other studies, the Healthy Ageing Quiz [19] and Elderly Awareness on Healthy Lifestyle during Ageing [17] were valid and reliable; however, the focus was again specifically on healthy ageing and did not reflect active ageing as a whole. Therefore, there is a need to develop a valid and reliable measurement tool that fully represents the active ageing concept.

The initial step in designing an Active Ageing Awareness Questionnaire (AAAQ) involved developing the three pillars that represent the WHO’s Active Ageing Framework, which are health, participation, and security. These three pillars served as a guide in the form of constructs for the development and analysis of the items. As the efficiency of interventions decreases with advancing age, interventions are more effective in early and middle adulthood [20]. This functions as an investment towards active ageing later in life and in turn, enhances the longevity and quality of life. The AAAQ is expected to measure the baseline population’s awareness of active ageing that will be instrumental for the development of an evidence-informed active ageing programme in Malaysia.

The intention of the tool is to measure active ageing awareness in the middle-aged population. This tool is different from the existing tools available that mainly measure active ageing abilities and capacities among older people [21]. Active ageing awareness should be measured for the population that might have risk factors for non-communicable diseases but have not developed or have complications of the diseases. Therefore, this awareness should be made during this critical period before reaching older age.

## 2. Materials and Methods

A pilot test-retest and exploratory factor analysis (EFA) and confirmatory factor analysis (CFA) were conducted to test the structural validity of the AAAQ. The assessment was conducted in Kulaijaya and Kota Tinggi which are two districts in Johor, Malaysia. The participants were from a non-professional group of employees from the private and public sectors [22]. Simple randomisation sampling was employed to recruit the participants with the help of the selected organisations.

The inclusion criteria for the participants were being Malaysian (Malay, Chinese, or Indian) and aged between 40 and 60 years old. If the participants were non-consenting, were aged 39 years and younger or 61 years and older or were unable to read and understand the Malay language, they were excluded from the study. Ethical approval was obtained from the Ministry of Health, Malaysia and was approved by the Research Ethics Committee (NMRR–16-40-28747) on 7 April 2016. Informed consent was obtained from the participants before the start of the study. 

### 2.1. Content Validation

Content validation of the AAAQ was to ensure that the items were relevant for measuring the awareness of active ageing among the target population, specifically in the Malaysian context. Therefore, three experts in public health and one in gerontology and geriatrics were asked to review and evaluate the items and rate the relevance of each item on a four-point Likert scale. Overall, the AAAQ was calculated to have a content validity index for individual items (I-CVI) of 1.00 and a content validity index for scale (S-CVI) of 1.00. In line with the experts’ suggestions, a I-CVI ≥ 0.78 and a S-CVI ≥ 0.9 from at least three experts were considered to have good content validity [23]. Thus, the initial AAAQ contained two stand-alone questions and 22 items. The original questionnaire and reasons for exclusion can be viewed in Appendix A as part of the Appendix A.

### 2.2. Linguistic Validation

The initial questionnaire was developed in English. The English version was sent for forward and backward translation to two native Malay speakers who had a medical background and who were proficient in English. The first person translated the questionnaire into Malay, the local and official language of Malaysia. After that, the second person translated the Malay version back into English and finally compared it with the initial questionnaire from the researchers.

### 2.3. Face Validation

The face validity check was done by recruiting five participants aged between 40 and 60 years old who were working in a public organisation and were classified as belonging to a non-professional group of employees. They were asked to answer the questionnaire and give feedback on the items in the questionnaire: the wording and understanding of the questions, the length of the questionnaire, the scoring system used, and the time needed to complete the questionnaire. The feedback from the participants was reviewed and taken into account but was not included in the data analysis.

### 2.4. Psychometric Assessment

The psychometric assessment of the AAAQ involved a pilot test and a test-retest as well as validity and internal reliability testing using EFA and CFA. Data collection for the psychometric assessment of the AAAQ was conducted from October 2016 to January 2017. The sample size calculation consisted of the pilot, test-retest, EFA, and CFA. For the pilot test, as there were two observations and 22 items targeting an ICC of 0.9, a power of 80%, and five subjects, the calculated sample size was 110. For test-retest, a minimum of 50 participants was required [24]. For the EFA and CFA, an item and sample ratio was used whereby 1:5 and 1:10 were used for EFA and CFA, respectively [25,26]. With a dropout rate of 10%, the calculated sample size was 363 for both tests.

A total of 110 participants completed the pilot test, and 81 participants followed up for the retest two weeks later. A total of 404 participants were then recruited and completed the EFA and CFA. The eligible participants completed the EFA and CFA and later returned the questionnaires to the researchers. The data was entered into REDCap software (Vanderbilt University, Nashville, TN, USA) by two people, and then the researcher compared both sets of input data and cross-checked them with the questionnaire for any discrepancies. The researcher then verified the final dataset once again before exporting it to selected statistical software for further analysis. 

### 2.5. Data Analysis

The Statistical Package for the Social Science (SPSS) Version 23 (IBM, Armonk, NY, USA) and the Analysis of a Moment Structures (AMOS) Version 22 (IBM, Armonk, NY, USA) software programs were used. For the pilot test, mean values of each item were calculated and ensured that the highest correlation between each item and other items in the same construct ranged between 0.3 and 0.9. For the questionnaire to have reliability, the lowest correlated item-total correlation (CITC) in each construct must be more than 0.3 [27], and the Cronbach’s alpha value should be more than 0.7 [23]. For the test-retest, the ICC was calculated by comparing the item scores of the data from the two questionnaires. The ICC was classified as poor (<0.4), fair to good (0.4 to <0.75), and excellent (≥0.75) [28].

The purpose of using EFA was to try to identify whether the items in the questionnaire were suitable for structure detection. Bartlett’s test of sphericity was used to test the items’ correlation matrix, while the Kaiser–Meyer–Olkin (KMO) was used to measure sampling adequacy. A Bartlett’s test of sphericity with a significance level of less than 0.5 and a KMO of more than 0.7 meant that the items were suitable for structure detection and factor analysis [29]. After conducting EFA by using principal axis factoring, the eigenvalue of the factors extracted was compared with that of a parallel EFA analysis to confirm the number of factors that remained in the EFA [30] and ensure that the eigenvalue was more than 1 [27].

In a CFA, a tool undergoes assessments for dimensionality, validity, and reliability [31] in which dimensionality is achieved when all the items have a factor loading of more than 0.5 [26,27,31]. In addition, the questionnaire was tested for three types of validity: convergent, construct, and discriminant. Convergent validity exists when the average variance extracted (AVE) for each construct is ≥ 0.5, construct validity is achieved when the fitness indices of the model fulfil their criteria, and discriminant validity examines the redundancy of the constructs in the model [31]. Discriminant validity is achieved when the maximum shared variance (MSV) or the average shared variance (ASV) is less than the AVE [30]. The fitness of the model was also tested by using several indices that are commonly used in the literature, namely, chi-square/degrees of freedom (CMIN/df < 3) [32], goodness of fit index (GFI > 0.9) [33], comparative fit index (CFI > 0.9) [34], and root mean square of error approximation (RMSEA < 0.08) [35].

Finally, the reliability of the questionnaire was assessed to determine how strongly the measurement items held together in each construct. This was achieved when the Cronbach’s alpha for each construct exceeded 0.7 [27]. Composite reliability (CR), which indicates the reliability and internal consistency of the latent construct, was achieved when CR ≥ 0.6 for every construct. The AVE, which indicates the average percentage of variance explained by the latent construct, needed to be more than 0.5 [27,30].

## 3. Results

### 3.1. Characteristics of the Participants

Out of the 110 participants that were approached during the pilot test, only 81 (74%) responded during the test-retest (Table 1), while 404 participants responded for EFA and CFA tests.

### 3.2. Reliability Analysis

In the pilot test analysis, the items in the questionnaire were assessed according to their respective constructs; namely, health, participation, and security. The constructs had 11, 7, and 4 items, respectively. Table 2 shows the inter-item correlation for the three constructs. An inter-item correlation value of more than 0.9 indicates that an item addresses the same issue as one other item. It can be seen that none of the inter-item correlations have a value greater than 0.9 except for items 23 and 24, indicating that one of these items should be deleted due to the presence of redundancy.

The Cronbach’s alpha values for health (α = 0.892), participation (α = 0.800), and security (α = 0.931) were at a value of more than 0.7. Both health (CITC = 0.296) and participation (CITC = 0.271) constructs had a corrected item-total correlation (CITC) of less than 0.3, while security was at 0.711. The CITC measures the reliability of a multi-item scale.

The results of the test-retest analysis are shown in Table 3. It can be seen that the ICCs for all items range from 0.489 for item A6 to 0.788 for item A7. The preferred ICC value was > 0.75, which was achieved by items A7, A14, A18, and A22. The ICC values of the other items ranged from 0.489 to 0.739, which were considered “fair to good” [28]. Thus, the findings of the test-retest confirmed the stability of the AAAQ.

### 3.3. Exploratory Factor Analysis (EFA)

The initial analysis showed that the four factors that were extracted explained 67 percent of the variance in the 22 items of the AAAQ. However, it was noticed that one of the eigenvalues was less than 1 (factor 1 = 10.971, factor 2 = 1.860, factor 3 = 1.077, factor 4 = 0.633), which indicated that the factor was not significant [27]. Therefore, in order to determine the ideal number of factors that should be extracted for the AAAQ, a parallel analysis was conducted [30], and the eigenvalues of both analyses were compared. The parallel analysis (factor 1 = 1.7422, factor 2 = 1.6289, factor 3 = 1.5289, factor 4 = 1.4312) showed all eigenvalues had a value of more than 1. However, factors 3 and 4 were removed, as the eigenvalues were either too close or had less than a value of 1.

Subsequently, the EFA was repeated, but the number of factors was fixed at two. The findings showed that the two factors explained 58 percent of the variance in the 22 items of the AAAQ. In addition, the KMO measure of sampling adequacy for the two factors was 0.916 at a significance level of less than 0.001, which was considered good [27]. However, it was observed that items A23 and A24 were highly correlated with each other, i.e., the correlation coefficient value was more than 0.9, which suggested that the redundancy should be removed. Table 4 shows the distribution of the items among the two factors. It can be seen that the first factor consists of all the items that belong to the health construct while the items in the second factor were previously assigned to the participation and security constructs. Therefore, prior to conducting CFA, the two factors were renamed as the health and non-health constructs.

### 3.4. Confirmatory Factor Analysis (CFA)

In the CFA, the 22 items were analysed according to the two constructs mentioned above. The model was analysed step by step until the fitness of the model was confirmed. All 22 items in Model 1 were included in the first modelling analysis (Figure 1). The fitness of the model was not achieved by Model 1 because 2 of the 22 items in Model 1 were highly correlated with each other as identified in the EFA. The items in question were items A23 and A24. Of these two items, item A24 had a lower factor loading (0.956) as compared to item A23 (0.974).

In light of the above, before running the second model, item A24 was deleted. Moreover, four items with a factor loading less than 0.5, namely, A4, A14, A17, and A18, were also deleted because items with a factor loading below this value could also affect the fitness of the model [31]. After removing the five items in Model 2, the unidimensionality of the model was achieved, however, the fitness of the model was not achieved (Table 5).

After the second model failed to achieve model fitness, the modification indices (MI) were examined. A MI value of more than 15 suggested that items were redundant [31]. Therefore, at this point in the CFA, items with a high MI were deleted, and some of the paired items were set as free parameters. Therefore, the MI was re-evaluated after Model 3, and another pair of items were set as free parameters. Finally, the fitness of the fourth model was achieved. The items that were deleted in Model 2 and Model 3 were items A7, A12, and A23, while the paired items of A3–A5, A6–A8, A19–A20 and A21–A22 were set as free parameters. In Model 4, the construct validity was acceptable (Figure 2). The findings in Table 6 were obtained after the fitness of the model was achieved in Model 4. The final model consists of 14 items: 8 items from the health construct and 6 items from the non-health construct.

Finally, the composite reliabilities (CR) for the health and non-health constructs were 0.91 and 0.89, respectively, while the average variances extracted (AVE) were 0.56 and 0.58, respectively. Furthermore, the Cronbach’s alphas for both constructs were 0.92 and 0.91, respectively. These findings indicated that the 14 items remaining in the final model of the AAAQ achieved satisfactory reliability [27]. The AAAQ also achieved convergent validity for the AVE for both constructs, which was more than 0.5 [27].

## 4. Discussion

The development of the AAAQ was conducted systematically, and its psychometric properties were evaluated by testing it on a sample of 404 middle-aged persons (EFA and CFA only). The development of the AAAQ was guided by published literature and the Active Ageing Framework endorsed by the WHO [10] as well as input and feedback from experts in public health, geriatrics, and gerontology. The AAAQ tool was intended to assess whether participants were aware of the three pillars of active ageing.

Most of the available tools have been developed for a specific population and set of objectives, which restricts usability for the general population and means that they are not appropriate for comparison. For example, the active ageing measurement tool developed by Perales et al. [36] targeted European countries, while Kattika et al. [37] developed the tool to assess active ageing attributes based on Thai culture. In addition, the evaluations of the related available tools to measure awareness of ageing show limitations in terms of their validity and reliability as well as the issue being measured [18,38]. Moreover, some of them have not completely incorporated all three pillars of active ageing, namely, health, participation, and security, into the questionnaire [17,19]. Furthermore, the validation of these tools is not fully explained, and none of them have been translated and validated for local users. Therefore, the authors became interested in developing a tool to measure the awareness of active ageing among Malaysians.

EFA is a statistical technique that is appropriate for scale development and is used to test or measure the underlying theory for hypothesised patterns of loading [39]. During the development of the items, initially, three factors were developed to represent the three pillars of the Active Ageing Framework. However, it became apparent from the EFA results that the three pillars were inextricable. The EFA yielded two factors for the 22 items that explained 58 percent of the variance in the items at a significance level of less than 0.001, which indicated a good explanatory power. It was apparent from the EFA results that all the items appropriately belonged to these two respective factors, and these two factors were renamed health and non-health factors. The EFA also revealed that two items were highly correlated with each other, and it was noted that this issue might affect the model fit later.

CFA is a statistical technique that is used to verify the factor structure in a set of items. It allows the researcher to test the hypothesis that there is a relationship between the items and the factors to which they belong [40]. Apart from factor loading, convergent validity in this study was examined by observing the CR and AVE for each factor in the AAAQ. The values of CR and AVE should be more than 0.5 and 0.6, respectively. A lower CR indicates that the items do not measure what they are intended to measure. A low AVE indicates that more errors remain in the items than the variance explained by the intended factor [27].

In the CFA, the initial 22 items were loaded into two factors (health and non-health). Then, a repeated process of modification was performed based on the factor loading of each item and the correlation between the items and factors as well as the model fit. Initially, all the items with a factor loading of less than 0.4 were deleted. In addition, when two items were highly correlated, the one with the lower factor loading was deleted. The MI were also examined, and when there were pairs of items that had high values, one of them was deleted or they were set as free parameters. Finally, the model with the best fit based on RMSEA, GFI, CFI, TLI, and x^2^/df was kept. The findings suggested that the model with 14 items had the best fit and achieved construct validity, which indicated that the AAAQ was able to distinguish between those who were aware of active ageing and those who were not. Thus, in the final model, 8 out of the 22 items were deleted (36 percent). It has been suggested that the proportion of items deleted should not be more than 20 percent [31].

The deleted items were essential factors for active ageing as agreed upon by experts, but the psychometric assessments among the study population did not support this view. Most probably, the item statements were not well understood by the participants. Furthermore, the development of the items was based on the findings and opinions of older persons, and the participants, who were in a younger age group, might not agree with the items until they became elderly. The CR, AVE, and standardised factor loading of the final model with 14 items indicated that convergent validity was achieved. The two structures in this scale were considered good because all the factor loadings of the items were more than 0.5, the AVEs were more than 0.5, and the CRs were more than 0.6. This result indicated that there was sufficient convergent validity in this tool. Thus, the items were well correlated with their respective factors. Cronbach’s alpha, AVE, and CR values confirmed that the AAAQ had internal consistency.

On the other hand, the AAAQ did not achieve discriminant validity. The maximum shared variance (MSV) and average shared variance (ASV) for both the health and non-health constructs were found to be 0.86. Thus, the values of both the MSV and ASV were higher than that of the AVE of the constructs, which suggested that discriminant validity was not achieved. However, even though discriminant validity was not achieved by the AAAQ, the overall validity of the AAAQ was not affected. In the AAAQ, all 14 items covering both constructs were summed up to get a score for the awareness of active ageing as there was no intention to discriminate awareness of active ageing between the two constructs. This is because the review of the literature confirmed that the pillars of the Active Ageing Framework are inextricable. The fact that the three pillars of the Active Ageing Framework are inextricable may be the cause for the lack of discriminant validity [41,42].

This study has a number of limitations. Firstly, the result is based on self-reported measures, which are a type of measurement that is prone to response and information bias. Secondly, the psychometric assessment of the AAAQ involved only the Malay version. It may be quite challenging to validate the English version in this study population, as English is not their main language. Furthermore, the authors did not measure any other health-related or ageing-related constructs to correlate with the AAAQ. As this is one of the first instruments to incorporate the Active Ageing Framework, there is no gold standard against which to evaluate its criterion validity. Thus, criterion validation of the AAAQ cannot be established. In addition, the category of ICC defined as “fair to good” may be not enough to use generally. In a newly constructed tool, although a thorough literature search was done, there is still the possibility of flaws in the tools, especially when constructing items for the domain. It also relies on the understanding of the respondents for each item, and thus future research is needed to improve on the tool. Finally, the sample is fairly representative of the particular age group of interest, but not all of the middle-aged population in Malaysia, due to the limited sample.

In short, although the outcome of the EFA and CFA yielded only two factors, namely health and non-health factors, the essence of the active ageing framework is still captured in this questionnaire as it considers all three pillars, namely health, participation, and security. Therefore, the 16 items (2 stand-alone questions and 14 statements) of the Malay version of the AAAQ are satisfactorily reliable and valid, so the AAAQ can be used as a measurement tool to assess awareness of active ageing.

## 5. Conclusions

In summary, the relevancy of the AAAQ was tested by conducting content validation and face validity, reliability, and consistency evaluations as well as construct, convergent, and discriminant validity tests. From the results, it was confirmed that the Malay version of the AAAQ was a satisfactorily valid and reliable questionnaire with acceptable internal consistency that could be used in a local setting in Malaysia. Thus, it was suitable for use in measuring the awareness of active ageing among the middle-aged population in Malaysia, specifically in the state of Johor. Due to some limitations, care must be taken when interpreting the results of this questionnaire. This validated tool will enhance future public health research in this domain.

## Figures and Tables

**Figure 1 healthcare-09-00499-f001:**
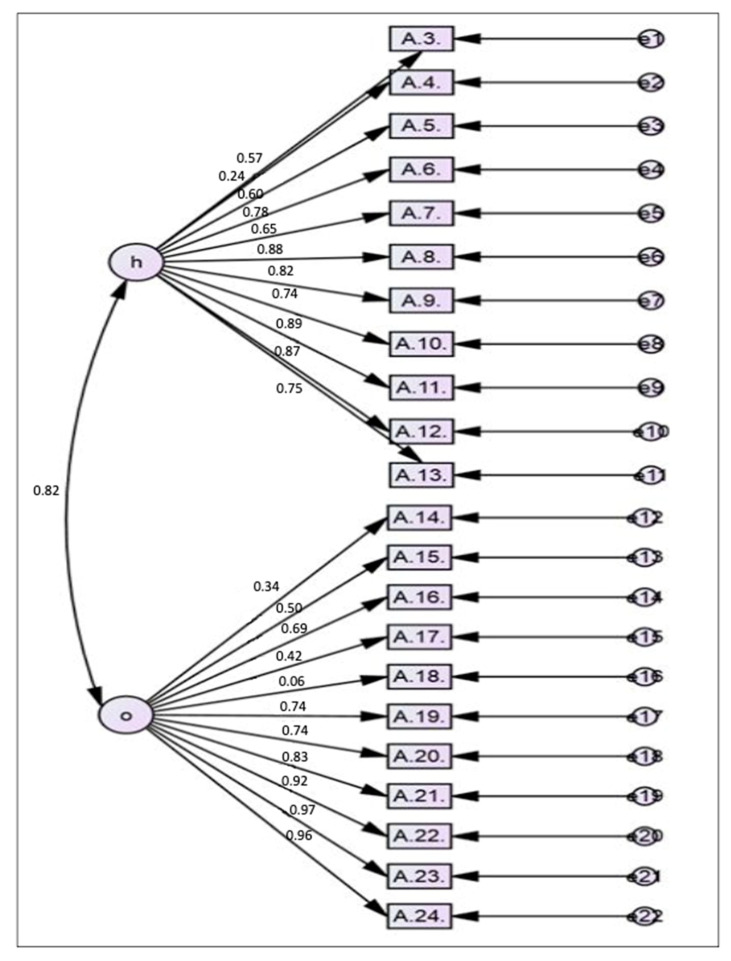
First model of the AAAQ containing 22 items.

**Figure 2 healthcare-09-00499-f002:**
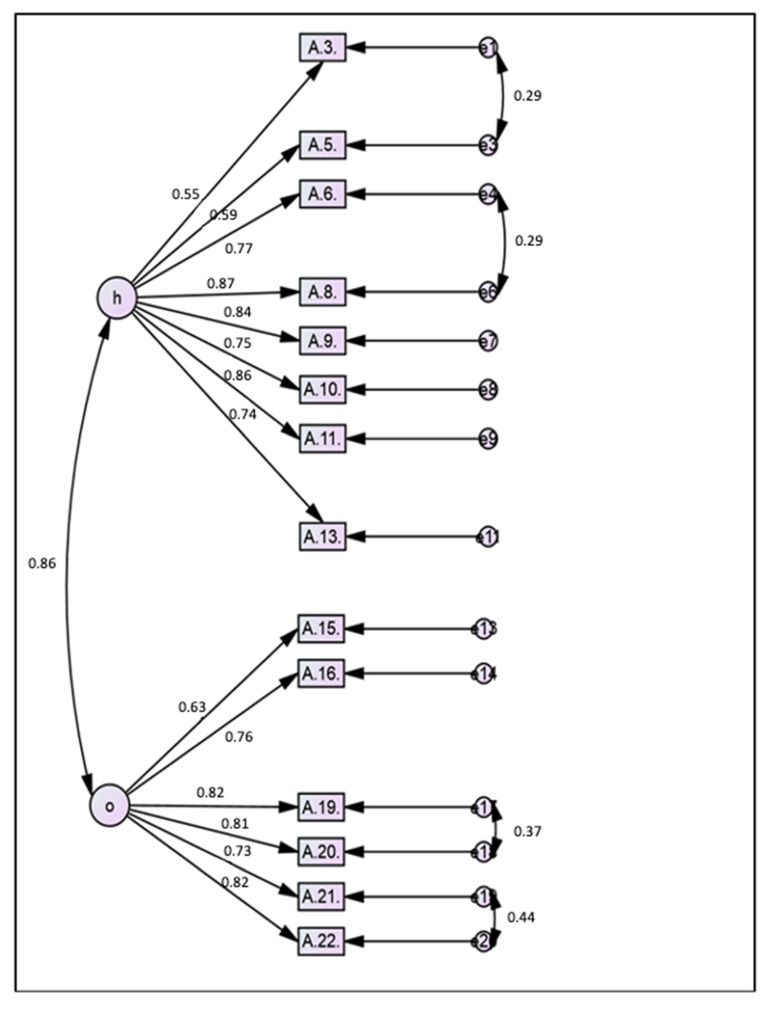
Final model of the AAAQ containing 14 items.

**Table 1 healthcare-09-00499-t001:** Characteristics of the participants for the AAAQ validation.

Characteristic	Pilot Test (*n* = 110)		Test-Retest (*n* = 81)		EFA/CFA (*n* = 404)	
	*n*	%	*n*	%	*n*	%
**Age (Years ± SD**)	50.19 ± 5.52		49.4 ± 5.7		49.9 ± 5.8	
*Gender:*						
Male	72	65.5	47	58.0	208	51.5
Female	38	34.5	34	42.0	196	48.5
*Ethnicity:*						
Malay	93	84.5	73	90.1	372	92.5
Chinese	9	8.2	5	6.2	18	4.5
Indian	8	7.3	3	3.7	12	3.0

**Table 2 healthcare-09-00499-t002:** The Inter-item Correlation for the constructs in the pilot test (n = 110).

Health Construct												Participation Construct							Security Construct			
	3	4	5	6	7	8	9	10	11	12	13	14	15	16	17	18	19	20	21	22	23	24
3	-	0.552	0.432	0.452	0.396	0.493	0.381	0.315	0.323	0.326	0.288											
4	0.552	-	0.317	0.203	0.226	0.236	0.070	0.095	0.258	0.181	0.033											
5	0.432	0.317	-	0.316	0.570	0.377	0.341	0.382	0.337	0.379	0.372											
6	0.452	0.203	0.316	-	0.501	0.811	0.580	0.418	0.512	0.442	0.389											
7	0.396	0.226	0.570	0.501	-	0.578	0.575	0.397	0.326	0.280	0.346											
8	0.493	0.236	0.377	0.811	0.578	-	0.588	0.491	0.660	0.619	0.435											
9	0.381	0.070	0.341	0.580	0.575	0.588	-	0.612	0.516	0.407	0.553											
10	0.315	0.095	0.382	0.418	0.397	0.491	0.612	-	0.550	0.557	0.594											
11	0.323	0.258	0.337	0.512	0.326	0.660	0.516	0.550	-	0.867	0.610											
12	0.326	0.181	0.379	0.442	0.280	0.619	0.407	0.557	0.867	-	0.708											
13	0.288	0.033	0.372	0.389	0.346	0.435	0.553	0.594	0.610	0.708	-											
14												-	0.286	0.162	0.387	0.220	0.215	0.208				
15												0.286	-	0.566	0.569	0.149	0.570	0.515				
16												0.162	0.566	-	0.520	0.214	0.625	0.558				
17												0.387	0.569	0.520	-	0.410	0.588	0.453				
18												0.220	0.149	0.214	0.410	-	0.179	0.045				
19												0.215	0.570	0.625	0.588	0.179	-	0.804				
20												0.208	0.515	0.558	0.453	0.045	0.804	-				
21																			-	0.670	0.720	0.661
22																			0.670	-	0.891	0.843
23																			0.720	0.891	-	0.920
24																			0.661	0.843	0.920	-

**Table 3 healthcare-09-00499-t003:** The intra-class correlation coefficients of the items in the test-retest (*n* = 81).

Dimension	Item	ICC	95% CI	*p*-Value
Health	A3	0.634	0.337–0.787	<0.001
	A4	0.628	0.424–0.760	<0.001
	A5	0.696	0.523–0.806	<0.001
	A6	0.486	0.209–0.667	<0.001
	A7	0.788	0.671–0.864	<0.001
	A8	0.624	0.414–0.758	<0.001
	A9	0.722	0.567–0.821	<0.001
	A10	0.712	0.554–0.814	<0.001
	A11	0.690	0.517–0.801	<0.001
	A12	0.584	0.351–0.733	<0.001
	A13	0.677	0.501–0.792	<0.001
Participation	A14	0.751	0.606–0.842	< 0.001
	A15	0.629	0.421–0.761	< 0.001
	A16	0.737	0.590–0.831	< 0.001
	A17	0.739	0.593–0.832	< 0.001
	A18	0.754	0.619–0.842	< 0.001
	A19	0.728	0.575–0.825	< 0.001
	A20	0.608	0.397–0.747	< 0.001
Security	A21	0.540	0.291–0.702	< 0.001
	A22	0.787	0.670–0.863	< 0.001
	A23	0.691	0.520–0.801	< 0.001
	A24	0.699	0.531–0.807	< 0.001

ICC: intra-class correlation coefficient; CI: confidence interval.

**Table 4 healthcare-09-00499-t004:** Pattern matrix of the AAAQ with two factors.

Item	Factor 1	Item	Factor 2
A3	0.586	A14	0.340
A4	0.239	A15	0.498
A5	0.601	A16	0.687
A6	0.779	A17	0.417
A7	0.654	A18	0.064
A8	0.882	A19	0.742
A9	0.821	A20	0.742
A10	0.738	A21	0.825
A11	0.886	A22	0.923
A12	0.867	A23	0.974
A13	0.754	A24	0.956

**Table 5 healthcare-09-00499-t005:** Fitness of the model for AAAQ.

Model	CMIN/DF	GFI	AGFI	TLI	CFI	RMSEA
<3	>0.9	>0.9	>0.9	>0.9	<0.08
1	4.448	0.772	0.717	0.853	0.870	0.107
2	5.362	0.750	0.676	0.840	0.861	0.131
3	3.150	0.884	0.833	0.922	0.938	0.092
4	2.771	0.903	0.859	0.951	0.938	0.082

CMIN/DF: chi-square/degree of freedom; GFI: goodness of fit; AGFI: average goodness of fit index; TLI: Tucker Lewis index; CFI: comparative fit index; RMSEA: root mean square of error approximation.

**Table 6 healthcare-09-00499-t006:** The CFA validity and reliability results for the final model.

Construct	Item	Factor Loading	Cronbach’s Alpha (≥0.7)	CR (≥0.6)	AVE (≥0.5)	Health Construct	Non-Health Construct
**Health**	3	0.55	0.92	0.91	0.56	0.75	-
	5	0.59					
	6	0.77					
	8	0.87					
	9	0.84					
	10	0.75					
	11	0.86					
	13	0.74					
**Non-health**	15	0.63	0.91	0.89	0.58	0.86	0.76
	16	0.76					
	19	0.82					
	20	0.81					
	21	0.73					
	22	0.82					

CR: composite reliability; AVE: average variance extracted.

## Data Availability

The datasets used and/or analysed during the current study are available from the corresponding author upon reasonable request.

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
