# Peer review of "Development of the Active Ageing Awareness Questionnaire in Malaysia"

_healthcare, 2021, doi:10.3390/healthcare9050499_

Round 1

Reviewer 1 Report

Thank you for giving me the opportunity to review the article. The author conducted the study focusing on the development of the AAAQ in Malaysia. The topic is socially important, but there are several methodological problems. Therefore, the reviewer thought that the manuscript should be revised before further considerations. I left comments below.

Comments:

Introduction:

  1. The authors should update the referenced article in the introduction section. Several articles are published over 10 years ago.
  2. The authors focused on the increase of aged population and its problems. However, they conducted the study only in the middle-aged population. Why did the authors not conduct the study in the aged population? The context should be describe in the Introduction and the Discussion sections.

Materials and Methods:

  1. The authors should add the ethical approval number and the date.
  2. Did the authors obtain the informed consent from the participants of this study? They should declare it in the Methods section.
  3. How to determine the number of participants included in each step of this study? The authors should describe it based on the calculation.
  4. The authors should provide the AAAQ Malay version which developed in this study as supplementary material. It should be valuable for potential readers who want to use it in their research.

Results:

  1. Did the authors not collect the other socioeconomic backgrounds of the participants?
  2. Why were the authors think that “fair to good” ICC value enough?

Discussion:

  1. Were the participants of this study representative for the middle-aged population in Malaysia? If not, the authors should discuss in detail as a limitation. Not only that, they should revise the conclusion statement according to this point.

Author Response

Introduction:

1. Added new citation in manuscript: (2019) - [20] (See attachment)

2. Added phrase: As the efficiency of interventions decreases with advancing age, interventions are more effective in early and middle adulthood.

Methodology

3. Added phrase: ethical approval number and date: (NMRR–16-40-28747) on 7th April 2016

4. Yes, added phrase: Informed consent were obtained from the participants before the start of the study.

5. Due to the word limit, the explanation was opted from the manuscript. It was conducted among the non-professional group of public employees aged between 40 and 60 years old with a sample size of 700 participants selected through simple random sampling from the sampling frame of eligible employees. Out of the total of 700 eligible participant, 595 completed the questionnaire (15% incomplete questionnaire or drop out).

6. Noted, the AAAQ will be optionally added as a supplementary material for the reader's reference

Results

7. Other socioeconomic background data have been collected such as household income, education level, occupation and even retirement plan. However, the authors feel that these information will be suited for a second manuscript once this validation paper is published.

8. ICC value fair to good: The reference for ICC score were based from: leiss, J.; Cohen, J. The Design and Analysis of Clinical Experiments; John Wiley & Sons: New York, United States, 1986.

Discussion

9. The participants were representative of middle-aged population in Malaysia.

Reviewer 2 Report

This is an interesting study developing and examining the psychometric properties of the Awareness of Active Ageing Questionnaire. The authors showed that the scale has satisfactory psychometric properties.

  1. In the introduction, it is necessary for the authors to define and elaborate what does it mean by "awareness of active ageing," While active ageing is an important issue, the justification of the development of this particular scale is lacking.
  2. It is necessary for the authors to report each item of AAAQ in the manuscript. It is not possible to evaluate the scale without knowing the items. This is also important so that the scale can be widely used.
  3. I don't think the authors examine the convergent and divergent validity of the scale accurately. The authors did not measure any other health-related or ageing-related constructs to correlate with AAAQ. It is a limitation that should be addressed

Author Response

  1. Added phrase in introduction: "Awareness of active ageing is operationally defined as a piece of knowledge about the determinants of active ageing." As discussed in the manuscript, there is currently no specific and standardised tool available to measure awareness of active ageing that combines the three pillars of active ageing effectively (i.e. health, participation and security). Several studies have been conducted on awareness of ageing in many settings internationally, however the aspect of ‘participation’ and ‘security’ dimensions were left out in the questionnaires.
  2. The AAAQ and its items shall be added as a supplementary material for the reader's reference.
  3. As described in the manuscript, convergent validity in this study was examined by observing the Composite Reliability and Average Variance Extracted for each factor in the AAAQ. Convergent validity exists when the average variance extracted (AVE) for each construct is ≥ 0.5; construct validity is achieved when the fitness indexes of the model fulfil their criteria. With that being said, the authors did not measure any other health-related or ageing-related constructs to correlate with AAAQ was added as a limitation.

Thank you for your comments. We have made the changes requested.

Reviewer 3 Report

Thank you for recommending me as a reviewer. This paper was to conducted the Awareness of Active Ageing Questionnaire (AAAQ) for content, linguistic and face validations, and test-retest reliability. Exploratory Factor Analysis (EFA) and Confirmatory Factor Analysis (CFA) were performed to test the structural validity of the AAAQ. If the authors complete the revision, the quality of the study will be further improved.

  1. abstract: "A total of 110 participants (mean ± SD = 50.19± 5.52) were selected for the pilot, 81 participants (mean ± SD = 49.40 ± 5.70) for the test-retest and 404 participants (mean ± SD = 49.90 ± 5.80) for CFA and EFA tests." - What do the mean and standard deviation mean in this sentence? Is it a age? Or is it a test score? If the author explains the abstract more specifically, it can help readers understand.

2. introduction: The introductory section is well written. However, some sentences are lengthy and difficult for readers to understand. If the author separates long sentences in short, it can help readers understand.

3. page 3: The description of the subject is insufficient. The author should describe the subject in more detail (ex. sampling, exclusion criteria, inclusion criteria, missing values, etc.

4. The conclusion section is well written, but it is too assertive. Research involves several limitations, so the expression of the conclusion needs to be change.

Author Response

  1. Added phrase: "mean age" in abstract
  2. Rephrased lengthy sentences in the introduction to make it easier for readers to understand. (see attachment)
  3. Added more information of the participants in method (see attachment). Due to word limit, the explanation of missing value was omitted from this manuscript.
  4. Expression of the conclusion has been tampered to manage expectations due to limitations. (see attachment)

Thank you for your comments. We have made the changes requested.

Round 2

Reviewer 1 Report

Thank you for giving me the opportunity to review the article. The author revised the manuscript partially. However, major points are not addressed in the current manuscript. Therefore, the reviewer thought that the manuscript cannot be accepted for publication. I left comments below.

AR, author’s response; AC, additional comment

Comments:

Introduction:

  1. The authors should update the referenced article in the introduction section. Several articles are published over 10 years ago.

AR: Added new citation in manuscript: (2019) - [20] (See attachment)

AC: The authors only added one citation. They should search literature carefully, and update the section.

  1. The authors focused on the increase of aged population and its problems. However, they conducted the study only in the middle-aged population. Why did the authors not conduct the study in the aged population? The context should be described in the Introduction and the Discussion sections.

AR: Added phrase: As the efficiency of interventions decreases with advancing age, interventions are more effective in early and middle adulthood.

AC: The authors revised the point appropriately.

Materials and Methods:

  1. The authors should add the ethical approval number and the date.

AR: Added phrase: ethical approval number and date: (NMRR–16-40-28747) on 7th April 2016

AC: The authors revised the point appropriately.

  1. Did the authors obtain the informed consent from the participants of this study? They should declare it in the Methods section.

AR: Yes, added phrase: Informed consent were obtained from the participants before the start of the study.

AC: The authors revised the point appropriately.

  1. How to determine the number of participants included in each step of this study? The authors should describe it based on the calculation.

AR: Due to the word limit, the explanation was opted from the manuscript. It was conducted among the non-professional group of public employees aged between 40 and 60 years old with a sample size of 700 participants selected through simple random sampling from the sampling frame of eligible employees. Out of the total of 700 eligible participant, 595 completed the questionnaire (15% incomplete questionnaire or drop out).

AC: There is no upper limit to the number of words (“a main text of around 3000 words at minimum”). The authors should read the instructions for authors, and include important information of their study. In addition to this, the response is inappropriate to describe about the sample size calculation.

  1. The authors should provide the AAAQ Malay version which developed in this study as supplementary material. It should be valuable for potential readers who want to use it in their research.

AR: Noted, the AAAQ will be optionally added as a supplementary material for the reader's reference

AC: The authors revised the point appropriately.

Results:

  1. Did the authors not collect the other socioeconomic backgrounds of the participants?

AR: Other socioeconomic background data have been collected such as household income, education level, occupation and even retirement plan. However, the authors feel that these information will be suited for a second manuscript once this validation paper is published.

AC: There is a crucial problem with the answer. Why did the authors collect the socioeconomic background when they design the study? Why they think to split the study arbitrarily? All of the authors should discuss about the problem deeply, and the reviewer expect authors to be honest in their writing.

  1. Why were the authors think that “fair to good” ICC value enough?

AR: ICC value fair to good: The reference for ICC score were based from: leiss, J.; Cohen, J. The Design and Analysis of Clinical Experiments; John Wiley & Sons: New York, United States, 1986.

AC: The category is “fair to good”, and it may be not enough to use generally. The authors should read the related articles, and deepen your understanding of this parameter.

Discussion:

  1. Were the participants of this study representative for the middle-aged population in Malaysia? If not, the authors should discuss in detail as a limitation. Not only that, they should revise the conclusion statement according to this point.

AR: The participants were representative of middle-aged population in Malaysia.

AC: The authors should describe why they think their population is representative appropriately. The authors should try to answer sincerely.

Author Response

Dear Reviewer,

Thank you for your comments. In light of your response, we understand your concerns and have taken major steps to revise our manuscript accordingly.

Kind regards.

---

1. AC: The authors only added one citation. They should search literature carefully, and update the section.

Revised and updated the references from [1-20] (Refer the attachment for list of changes)

5. AC: There is no upper limit to the number of words (“a main text of around 3000 words at minimum”). The authors should read the instructions for authors, and include important information of their study. In addition to this, the response is inappropriate to describe about the sample size calculation.

Changes made in 2.4 Psychometric Assessment (Pg.3) (Refer the attachment for wording changes)

Author 1:

There are several methods of sample size calculation for the validation study. We have consulted a statistician who is an expert in the validation study, and the method that we have chosen is also found in the majority of the published validation study.    

a. Bujang, M. A., & Baharum, N. (2017). A simplified guide to determination of sample size requirements for estimating the value of intraclass correlation coefficient: a review. In Archives of Orofacial Sciences The Journal of the School of Dental Sciences, USM Arch Orofac Sci (Vol. 12).

b. Andrew L. Comrey, & Howard B. Lee. (1992). A first course in factor analysis (Second ed.). New Jersey, Hillsdale: Lawrence Erlbaum Associates, Inc.

c. Nicholas D. Myers, S. A., and Ying Jin. (2011). Sample Size and Power Estimates for a Confirmatory Factor Analytic Model in Exercise and Sport: A Monte Carlo Approach. Research Quarterly for Exercise and Sport, 82(3), 412-423.

Author 2:

For the pilot-test, there are 2 observations and 22 items, targeting an ICC of 0.9, a power of 80% and having 5 subjects, the calculated sample size is 110. After consulting two statistician for sample size, the recommended sample size for test-retest is a minimum of 50 participants. 

Based on the work from Andrew, Comrey & Howard (1992) and Nicholas (2011), a ratio of 1:2 was the minimum and 1:10 was the maximum. A ratio of 1:10 was selected, so 22 items x 10 = 220 sample size required for CFA and EFA (minimum is 1:2 - 44 sample size). With a drop out rate of 10%, the sample is increased to 250. Combining both CFA and EFA is 500 participants.

As only 404 participants returned the questionnaire, the authors decided to split the sample size of 250 for CFA (1:10 ratio) and 154 for EFA (1:5 ratio).

7. AC: There is a crucial problem with the answer. Why did the authors collect the socioeconomic background when they design the study? Why they think to split the study arbitrarily? All of the authors should discuss about the problem deeply, and the reviewer expect authors to be honest in their writing.

The main focus of the questionnaire validation study is to test the reliability and validity of the newly developed study instrument. Thus, we only collect the basic socio-demographic information. By using the validated questionnaire, we further collected data on more comprehensive socio-demographic variables. The manuscript on the socio-economic difference of Active Ageing Awareness is in progress. This point has been agreed by the authors.

8. AC: The category for ICC is “fair to good”, and it may be not enough to use generally. The authors should read the related articles, and deepen your understanding of this parameter.

Changes: Added as limitation in 4. Discussion (Pg. 9) (Refer the attachment for wording changes)

Author 1:  

In a newly construct tool, although a thorough literature search was done, there is still possibility of flaws in the tools especially when constructing items for the domain. It also relies on the understanding of the respondents for each item. All these are the avenue for us and the future researchers to improve on the tool. 

Author 2:

As the analysis for ICC cannot be used alone, this questionnaire had gone through the whole validation process with different samples. The final analysis have removed several items based on the fitness.

9. AC: The authors should describe why they think their population is representative appropriately. The authors should try to answer sincerely.

Changes: The introduction (Pg. 1) conclusion and limitation (Pg 9-10) has been revised to include this point. (Refer the attachment for wording changes)

Author 1: Because of the nature of the questionnaire, we decided to include an older cohort of the middle-aged population (40-60 years). Our sample is fairly representative for the particular age group of interest but not all the middle aged population in Malaysia. 

Author 2: The intention of the tool is to measure active ageing awareness in the middle-aged population. This tool is different from the existing tools available, that mainly measure active ageing abilities and capacities among the older people. Active ageing awareness should be measured for the population that might have risk factors for non-communicable diseases (NCD), but have not developed the disease or have complications of the disease. This awareness should be made in this critical period of time, because they perceive that they are 'healthy' but they are actually 'a time bomb' if they continue to have the sedentary lifestyle.
